# Examination of PTSD and Depression Levels and Demographic Data of Syrian Refugee Children during the Pandemic

**Elif Erol** [1,*] **and Dilara Demirpençe Seçinti** [2]

1 Clinical Psychology Department, Halic Campus, Istanbul Rumeli University, Istanbul 34450, Turkey
2 Sisli Hamidiye Research and Training Hospital, Istanbul 34360, Turkey; dilarademirpence@gmail.com
* Correspondence: elifkoca5@yahoo.com

**Abstract:** Background: The worldwide population of child refugees is estimated to be over 10 million. Refugee children and adolescents are among the most vulnerable groups in the world, and the pandemic created new challenges for them. Objective: This study aimed to examine the PTSD and depression levels of Syrian refugee children and adolescents, the difficulties they experienced in access to food and education, and the changes in their family income, and evaluate the effects of these factors on symptom severities of depression and PTSD. Methods: We used data obtained from 631 Syrian refugee children between the ages of 7 and 15. Assessment measures for exposure to PTSD and depression included a socio-demographic form, stressors related to COVID-19, the Child and Adolescent Trauma Survey (CATS), and the patient-rated Children's Depression Inventory (CDI). ANCOVA is conducted to evaluate the differences between the symptoms of PTSD and depression. The regression analysis was used to determine the relationship between the scales and the demographic data. Results: The study finds that 40.7% of the refugee children experienced at least one trauma, 24.9% met the criteria for PTSD, and 15.5% met the criteria for depression. The changes in income and food access are associated with PTSD and depression, while access to education is not associated with PTSD and depression. The adolescents aged between 12 and 15 show more depression and PTSD symptoms then the children aged between 7 and 11. Conclusions: The study revealed that the most risky group in refugee children and adolescents in terms of PTSD and depression were the adolescents aged between 12 and 15 and the children and adolescent whose family income was decreased and had limited access to food.

**Keywords:** refugee; pandemic; refugee children and adolescents; mental health of refugee children

## 1. Introduction

On 31 December 2019, a novel coronavirus was first detected in Wuhan City, Hubei Province, China, and on 11 March 2020, the World Health Organization (WHO) declared the COVID-19 outbreak a global pandemic. The first COVID-19 case in Turkey was announced on 11 March 2020. A curfew was first imposed for citizens over the age of 65 and then for those under the age of 20. A nationwide curfew was announced later. With the declaration of a curfew for those under the age of 20, distance education started as of 17 March with the coordination of the Turkish Ministry of National Education and the Ministry of Health. While the broadcast of distance education programs started on a national channel with the Educational Informatics Network (EIN), a distance education platform, many private schools preferred to post content on their online learning platforms.

Refugees and replaced people are among the world's most vulnerable people affected by pandemic-related difficulties [1]. According to a report published by the Turkish Republic Ministry of Interior Directorate General for Migration Management (TRMIGM), the Syrian refugee population in Turkey is approximately 3.6 million, and children and adolescents represent 30% of this population [2]. Refugee children and adolescents are the most vulnerable group in terms of educational, social, economic, and psychological risk

factors, as they were obliged to resist the destructive effects of the war in their country, immigrate to a new country, and adapt to the host society and a new lifestyle [3]. The sources of stress experienced by refugee children and adolescents can be grouped into two categories: pre-migration and post-migration. While pre-migration factors include problems such as living in war zones, having inadequate means, and growing up under life threat; post-migration stressors include language barriers, exposure to discrimination, peer bullying, difficulties in access to education and health services, language barriers in education, adaptation to a new place, and uncertainty [4,5].

The most common psychiatric disorders among child and adolescent refugees are PTSD and depression. Meta-analytic studies showed that PTSD varied from 19% to 53% and depression from 10% to 33% [5–7]. Other studies in the literature reference emotional problems such as sadness, hopelessness, depressive mood, cognitive problems, fear, anger, and frustration [7]. The studies show an increase in psychiatric symptoms during the COVID-19 pandemic. The increased psychiatric symptoms are linked not only to fear of transmission, hospitalization, or the death of a loved one due to COVID-19, but also financial, educational, and social challenges. Due to lack of internet access, many school age children were deprived of education [2]. Therefore, it was hypothesized that COVID-19 related challenges such as financial problems or lack of access to education had effects on psychiatric symptoms.

Previous studies show that financial difficulties, fear of infection, unknown course of infection and its treatment, feelings of frustration as a result of the prolonged quarantine period, lack of information, difficulty in accessing basic needs, and inadequate medical service during quarantine caused negative psychological consequences such as depression and anxiety disorder [8,9]. Many adults have experienced job loss and socioeconomic adversities [10]. The literature shows that even the most developed states of the world encountered difficulties in fighting against the virus and maintaining their financial status and daily activities as before [11]. It is known that the most important determinant of the emotional stress experienced during the pandemic is derived from economic and psychosocial stressors (lifestyle and economic deterioration) and feelings of hopelessness before the pandemic [12]. The socioeconomic status (parents' income and education level) and access to school were associated with mental health of children and adolescents [13,14]. Thus, in this study, the effects of income and access to food as the economic indicators on depression and PTSD were investigated.

During the pandemic period, it is predicted that refugee adolescents had very limited access to education and health services. In addition, the negative effect of the COVID-19 pandemic on the economic status of refugee families exacerbated their vulnerability [15]. Hence, it was assumed that the pandemic was more challenging for the refugee population, especially children and adolescents. A study in Myanmar, a disadvantaged region, showed that 43.7% of the population had at least one psychiatric disorder [16].

Mental health studies conducted with refugee adolescents during the pandemic period are very limited in the literature. The objective of our study was to examine the effects of the COVID-19 pandemic on the mental health of refugee children in order to understand their needs and to ascertain depression and PTSD levels of children and adolescents in the 7–15 age group during the pandemic period and their differences regarding demographic variables such as age and access to health and education services.

*Key Practitioner Message*

What is known? Refugee children and adolescents are the group most vulnerable to educational, social, economic, and psychological risk factors. PTSD and depression are the most common psychiatric disorders observed among adolescent and child refugees.

What is new? The recent pandemic is a new major risk factor for refugee children and adolescents. It also made it very hard to contact them, both personally and through official channels.

What is significant to for clinical practice? It was found that the depression level for refugee adolescents relatively lower in Turkey. The reasons and results have been discussed in the paper.

## 2. Methods

### 2.1. Recruitment and Ethical Considerations

The research data were collected from children and adolescents who came to the Kumkapı European Side Coordination Centre with their families during April 2021. The Kumkapı European Side Coordination Centre is affiliated with the Istanbul Provincial Immigration Administration. Refugees visit the center in order to obtain temporary protection documents, travel permission and temporary residence permission, make applications for financial support provided by the government only to refugees, and update their legal documents. In Turkey, most of the Syrian refugee population is under temporary protection, so refugees have to visit this Centre with their family regularly [2].

### 2.2. Participants

The Syrian refugee families with children aged between 7 and 15 who visited the Centre were invited to the study by a staff working in the Centre. The criteria for participating in the study were to be between the ages of 7 to 15, to be a Syrian refugee living in Turkey, to have left Syria due to war between the years of 2011 and 2018, and proficiency in reading Turkish texts. Fifteen children who could not read Turkish texts and whose participation was not approved by their parents were excluded from the study. The participants were residents in Istanbul. In Turkey, most Syrian refugees are Muslim.

### 2.3. Measures

2.3.1. Sociodemographic Data Form

Data collected from the participants included gender, parents' level of education, participant's educational status, and access to psychological support.

2.3.2. Stressors Related to COVID-19

The stressors related to COVID-19 are defined as financial, educational, and social. It consists of 3 items. As the age range of the participants is wide, simple questions were chosen. In order to understand financial difficulties, 3 questions (item 1: "Has there been any change in your family income or job status?", item 2: "Has anyone in your family had difficulty getting her/his monthly salary or got fired due to COVID-19?", item 3: "How COVID-19 pandemic has affected your access to food?") were asked. The question investigating educational status during COVID-19 was "Do you have access to remote education?".

2.3.3. Child and Adolescent Trauma Screen (CATS) Youth Report

There is a limited number of DSM-5 based trauma questionnaires in Turkish, which we used for the Syrian refugee adolescents who could read Turkish. The permission for a Turkish form of CATS was obtained from Cedric Sachser.

CATS is a DSM-5 based scale that scans post-traumatic stress disorder symptoms in children and adolescents between the ages of 7 and 17 [17]. The Cronbach alpha value of CATS was found between 0.88 and 0.94 in studies conducted in different languages, and a structure with 4 sub-factors was determined using CFA.

The first stage of the scale consists of 15 items measuring potentially traumatic events (e.g., "Being around war", "Slapped, punched, or beat up in your family", and "Robbed by threat, force or weapon"). If the participant has any traumatic experience, they continue to the second part of the questionnaire. At the second stage, considering the exitance of the most stressful event, post-traumatic stress symptoms are examined. CATS is a Likert type scale consisting of 20 items (e.g., "Bad dreams reminding you of what happened", "Not wanting to do things you used to do", "Feeling mad", and "Having fits or anger and

taking it out on others"). Scoring 21 or above on the scale indicates clinically significant post-traumatic stress disorder symptoms.

The confirmatory factor analyses were conducted in order to show the validity of the questionnaire. The model exhibited good fit indices ($x^2$/SD = 4.34, RMSEA = 0.068, CFI = 0.91, GFI = 0.92) [18].

The Cronbach alpha value of the CATS scale was found to be 0.91 in this study. In addition, we calculated McDonald's omega to be 0.91.

### 2.3.4. Child Depression Inventory

The scale developed by [19] is a self-assessment scale used to measure the depression level in children. Analysis of its validity and reliability on a Turkish population was conducted by Oy (1991) [20], and the suggested cut-off point was 19 points. There are three responses for each item on the 27-item scale. The participant is asked to choose the statement that best describes themselves over the past two weeks. Each statement is given 0, 1, or 2 points according to the severity of symptom. The Cronbach value of CDI in this study is calculated as 0.72.

The confirmatory factor analyses were conducted in AMOS. The model exhibited good fit indices ($x^2$/SD = 3.15, RMSEA = 0.068, CFI = 0.71) [18].

### 2.4. Study Design

This is a cross sectional study.

### 2.5. Statistical Analysis

SPSS Statistic 24 package programs are used to analyze the data collected in the study. According to Tabachnick and Fidell [21], if the sample consisted of more than 200 participants, a parametric test can be used. Therefore, parametric tests were applied. The significance threshold was set at $p < 0.05$ for all analyses.

In the first phase of the study, the depression and post-traumatic stress disorder symptom scores of girls and boys between the ages of 7–11 and 12–15 years were compared to the *t* test results.

In the second phase of the study, regression analysis was performed to understand the effects of changes in family income, social isolation, access to online education, and difficulty in accessing food on depression and post-traumatic stress disorder scores. A food access scale is categorized in two levels; the first category covers those who had no problems in accessing food and those with mild difficulties, the second category covers those with moderate and severe lack of access to food. Regarding economic change, again, no change and a slight change in the level of income formed one category, whereas those with moderate and severe income loss formed another category.

## 3. Results

### 3.1. Socio-Demographic Features of General Participants

Participants in the study are refugees who migrated from Syria to Turkey between 2011 and 2018 and their average age is 10.08 ± 1.86. Of the 631 participants, 54.8% (*n* = 346) were boys and 45.2% (*n* = 285) were girls. While 76.2% (*n* = 481) of the participants were between the ages of 7 and 11, 23.8% (*n* = 150) were between the ages of 12 and 15. The sample reported that 6.2% worked before the pandemic, 25.8% did not go to school, and 98.3% did not receive any mental support (Table 1).

**Table 1.** Characteristics of the participants.

| Demographic Features | | Total | |
|---|---|---|---|
| | | N | % |
| Gender | Man | 346 | 54.8 |
| | Woman | 285 | 45.2 |
| Education level of mothers | Illiterate | 109 | 17.6 |
| | Primary school | 330 | 52.3 |
| | Middle School | 111 | 17.6 |
| | High school | 53 | 8.4 |
| | University and higher | 28 | 4.4 |
| Education level of fathers | Illiterate | 93 | 14.7 |
| | Primary school | 357 | 56.6 |
| | Middle School | 103 | 16.3 |
| | High school | 44 | 7 |
| | University and higher | 34 | 5.4 |
| Siblings | 0 | 5 | 0.8 |
| | 1–2 | 94 | 14.9 |
| | 3–4 | 287 | 45.5 |
| | 5 and more | 245 | 38.8 |
| Work situation | No | 39 | 6.2 |
| | Yes | 592 | 93.8 |
| Education of before Pandemic | No | 163 | 25.8 |
| | Yes | 468 | 74.2 |
| Psychological Support | No | 620 | 98.3 |
| | Yes | 11 | 1.7 |

*3.2. Stressors Related to COVID-19*

The study showed that 31.1% of participants could not receive any education, and 34.2% used online education platforms and watched lessons on the national TV channel during the COVID-19 pandemic. It was revealed that 3.5% had constant and severe difficulty in finding sufficient or quality (healthy) food. When questioned about their family income change, it was found that 8.1% had problems in covering basic needs and expenses. When the working status of their parents was questioned, 12.8% of reported that a family member quit his/her job and 61.8% preferred short term working and unpaid leave (Table 2).

*3.3. Mental Health Outcome of the Population*

The study showed that the rate of refugee children and adolescents who experienced at least one traumatic event was 40.7%. Of these children, 18.9% experienced a single trauma, 9.2% two traumas, 3.5% three traumas, and 9.1% more than three traumas. A total of 24.9% (*n* = 167) of the participants had clinically significant PTSD symptoms. In CDI, 9.7% of the sample population was reported to have clinically significant depression symptoms.

Depression and PTSD symptoms were examined for age categories (Table 3). It was revealed that adolescents aged between 12 and 15 showed significantly higher values than children aged between 7 and 11 in CDI ($t = -3.410$, $p < 0.01$) and CATS ($t = -2.726$, $p < 0.001$) questionnaires.

**Table 2.** Stressors related to COVID-19.

| Stressors | | N | % |
|---|---|---|---|
| Remote Education | Yes (I participated in all education) | 216 | 34.2 |
| | No (I have not received any education) | 196 | 31.1 |
| | Partially (I have access EBA TV but I could not have online access) | 219 | 34.7 |
| Change in income | No change | 157 | 24.9 |
| | Mild (There has been a mild change in income but there has not been any change in paying bills and providing essential needs) | 217 | 34.4 |
| | Medium (There has been some restrictions but there has not been any problems providing essential needs and paying bills) | 206 | 32.6 |
| | Serious (There has been problems providing essential needs) | 51 | 8.1 |
| Difficulties in getting salary due to pandemic | Yes (Left job) | 81 | 12.8 |
| | No (No problem) | 160 | 25.4 |
| | Partially (Unpaid leave) | 390 | 61.8 |
| Difficulties in accessing food | No change | 191 | 30.3 |
| | Mild (There has been enough food but there has been difficulties in accessing markets and finding necessities) | 205 | 32.5 |
| | Medium (There has been difficulties in accessing enough food intermittently) | 213 | 33.8 |
| | Serious (There has been always difficulties in accessing enough food) | 22 | 3.5 |
| Working in pandemic | No | 584 | 92.6 |
| | Yes | 47 | 7.4 |

**Table 3.** The Comparison of PTSD and Depressive Symptoms among Age Groups.

| | 7–11 (*n* = 481) | | 12–15 (*n* = 150) | | Total (*n* = 631) | | | |
|---|---|---|---|---|---|---|---|---|
| | Mean | SD | Mean | SD | Mean | SD | *t* | *p* |
| PTSD | 13.86 | 8.85 | 16.45 | 10.61 | 14.48 | 218 | −2.726 ** | 0.007 |
| Depressive Symptoms | 10.52 | 4.91 | 12.18 | 5.47 | 10.92 | 223 | −3.410 ** | 0.001 |

** *p* < 0.01.

### 3.4. The Association between Stressors Related to COVID-19 and Depression and PTSD

In the regression analysis section, six linear regression analyses were performed (shown in Table 4). The study examined food access, access to education, and loss of income factors during COVID-19 as predictors of depression and PTSD.

In the first analysis, in which depression was entered as the dependent variable, the moderate and severe change in the family's income level predicted the depression score ($R^2 = 0.023$, $F_{(3.627)} = 9.622$, $p < 0.001$). Moderate and severe stress in food access positively predicted depression scores ($R^2 = 0.018$, $F_{(3.627)} = 8.427$, $p < 0.01$), and negative changes in access to education services did not predict depression ($R^2 = 0.00$, $F_{(3.627)} = 4.548$, $p < 0.05$).

In the analysis where PTSD was entered as the dependent variable, changes in income level ($R^2 = 0.119$, $F_{(3.627)} = 8.632$, $p < 0.01$) and changes in food access positively predicted PTSD symptoms ($R^2 = 0.123$, $F_{(3.627)} = 8.860$, $p < 0.01$), while changes in access to education were not predictive of PTSD symptoms ($R^2 = -0.029$, $F_{(3.627)} = 5.686$, $p < 0.05$).

**Table 4.** The Relationship Sociodemographic Variables and Depressive Symptoms and PTSD.

| DV | IV | B | SE | $R^2$ | B | $t$ | $F$ |
|---|---|---|---|---|---|---|---|
| | Income | 1.537 | 0.387 | 0.023 | 0.155 *** | 4.104 | 9.622 |
| | Access to food | 1.334 | 0.394 | 0.018 | 0.133 ** | 3.989 | 8.427 |
| Depression | Access to education | 0.216 | 0.416 | 0.000 | 0.021 | 0.689 | 4.548 |
| | Income | 2.265 | 0.748 | 0.014 | 0.119 ** | 3.242 | 8.632 |
| PTSD | Access to food | 2.379 | 0.759 | 0.015 | 0.123 ** | 4.143 | 8.860 |
| | Access to education | −0.593 | 0.799 | 0.001 | −0.029 | −0.609 | 5.686 |

Adjusted for age and gender; ** $p < 0.01$; and *** $p < 0.01$.

## 4. Discussion

Refugees experienced traumatic events such as death, illness, and exposure to war; they experienced not only the trauma of living in a war zone, but they also faced challenges during their long and difficult travel from their homeland and difficult conditions in the host county [22]. COVID-19 caused new challenges for the refugee population. One of our hypotheses was that COVID-19 related challenges such as financial problems or lack of access to education had effects on psychiatric symptoms. It was supported by the findings. In our study, economic and educational difficulties experienced by 631 children and adolescents during the pandemic were investigated and their effects on PTSD and depression were examined. It was revealed that 40.7% of the sample population had experienced at least one traumatic event, and the rates of PTSD and depression were 24.7% and 9.7%, respectively. A total of 40.7% of the participants saw mild or severe changes to family income; 3.5% of them had difficulty in accessing food; 20.3% had no access to education; and only 1.7% of the population received support.

In our study, the PTSD prevalence and the number of the traumatic events experienced by the study population were in line with the studies that were conducted in Istanbul in 2019 [23] and with a systematic review study which shows the rate of PTSD was 11% [24]. On the other hand, a study conducted in Turkey with children and adolescents between the ages of 7 and 18 during the years of 2015 and 2016 showed that 45% of Syrian children and adolescents living in Istanbul met criteria of PTSD [25].

Another study conducted with the children of refugee families found that 79% of the children experienced at least one death in the family during war, migration, and travel, while 60% of them experienced or witnessed a loss or torture [26]. In our study, the number of traumatic events experienced by the population was lower compared to the previous studies; the difference may be related to the fact that the sample population were exposed to the war for a shorter period. The differences found in rates of PTSD between the studies may be connected to the use of different questionnaires and different data collection periods.

In this study, it was revealed that the depression rate of the population was in accordance with previous studies [26]. Additionally, as expected, adolescents aged between 12 and 15 had higher depressive symptoms and PTSD symptoms [27,28].

When the stressors related COVID-19 were examined, it was determined that 40.7% of the participants saw mild or severe changes to family income, and 3.5% of them had difficulty in accessing food; it was also found that income level and food access were predictors of depression and PTSD symptoms. In the adult refugee study, which was conducted during the pandemic period in April 2020 and included Syrian refugees living in Turkey, 41% of the refugee individuals stated that they lost their jobs as companies and enterprises were closed, while 18% lost their jobs due to the pandemic and 36% of employees lost some of their income [29].

Looking at the studies conducted during the pandemic period, it is observed that the most important factors affecting the mental health of adolescents and children are loss of family income, social isolation, previous psychiatric illness, age, and gender [15,30–33]. Although studies examining the effect of food access on adolescents' mental health during the pandemic period are limited in the literature, studies conducted before the pandemic have shown that the difficulty of food access is a determinant in terms

of depression and suicidal thoughts in adolescents [34]. Studies indicate that not only traumatic and ongoing negative life events, but also the presence of social risk factors can lead to psychopathologies such as post-traumatic stress disorders (PTSD), depression, and anxiety disorders [35]. Similar studies showed that the most determining factor in the development of psychological disorders during the 2003 SARS pandemic was the change in income level. It was found that PTSD and depression levels of children, whose families lost income and who had limited access to food during the pandemic period, increased [36]. In a study based on the effects of the COVID-19 period, it was found that anxiety symptoms were lower in adolescents with stable family income [8]. In another study, it was shown that the economic trauma which COVID-19 causes had an effect on PTSD and depression [37]. All these conclusions are in line with the findings of our study.

In our study, 25.8% of the children did not go to school before the pandemic; 40.2% of the children who went to school before the pandemic had full access to education, 36.8% had access to broadcast of education programs on the National Channel, and the remaining 20.3% had no access to education. However, educational access did not predict PTSD and depression. Looking at the literature on access to education, it is reported that 35.8% of Syrian school-age children living in Turkey do not go to school [38]. In a study conducted with adult refugees living in Turkey in April 2021, it was found that 70% of school-age children went to school, and 48% of school-age children could not access distance education due to lack of TV, tablet, and internet [29]. Another study found that disadvantaged groups had limited access to education during the pandemic due to socioeconomic factors [39]. Considering the effects of transition to remote education on child and adolescent mental health, it was revealed that school is a source of support, especially for children and adolescents with mental health problems [30]. And the school closure led to the worsening mental health of children whose family income was lower [40]. On the other hand, it was found that remote education has no direct effect on the depression rate of children and adolescents. However, it was revealed that online education had a negative impact on learning and the difficulties encountered during online education were associated with depression [41].

During the COVID-19 pandemic, the condition of children and adolescents with mental health problems has worsened and their sources of support have decreased [30]. On the other hand, concerning the effects of school closure and transition to online education on the mental health of children and adolescents, changes in access to educational services do not change the rate of depression in adolescents. However, learning difficulties due to transition to online education affected depression scores [41].

Similarly, our study shows that changes in access to education do not have a determinant effect on children's depression and PTSD.

Another finding of the study is that, despite the existing mental, social, financial, and educational difficulties, only 1.7% of the population received support. Such a low proportion of that high-risk population to receive support can be explained as follows: Individuals who have difficulties in accessing basic services are less likely to receive mental health support compared to the normal population. In addition, previous studies showed that the language barrier and fear of stigmatization reduce the applications for professional mental health support [42].

## 5. Limitations

The data in our study were collected from children and adolescents who live in Istanbul. For this reason, the lack of data of other refugee children and adolescents to the study may have limited the generalizability of the study. The fact that data in this study were collected using self-report questionnaires filled out by children between the ages of 7 and 12, lacking family forms, may present biases. However, studies conducted with this age group in the literature show that children provide information in parallel with their families [43]. The Cronbach's alpha coefficient of the Child Depression Inventory was low

for this sample group. On the other hand, although RMSEA and $\chi^2/df$ values of the model showed acceptable fit, CFI did not have a good fit [44,45].

## 6. Conclusions

Refugee children and adolescents are in the high-risk group for depression and PTSD during the COVID-19 pandemic. These children, especially those whose families have lost their income, who face difficulties in accessing food and education, and the 12 to 15 age group are at greater risk of mental difficulties. It was observed that only a very small number of these children received psychiatric support. Consequently, it was found that the population received little support during the COVID-19 pandemic period, despite increased mental, social, financial, and educational difficulties. For this reason, intervention plans should be developed for the higher institutions to review the living conditions of these children and adolescents and to provide the necessary educational and financial support to overcome the psychiatric difficulties they experienced.

**Author Contributions:** Conceptualization, investigation, resources, data curation, writing—original draft preparation, writing—review and editing, supervision, project administration, funding acquisition were contributed by E.E. and writing—review and editing, software, formal analysis were contributed by D.D.S. All authors have read and agreed to the published version of the manuscript.

**Funding:** This research was funded by Istanbul Rumeli University and Scientific Research Projects (Bilimsel Araştırma Projeleri—BAP), grant number BAP2021002.

**Institutional Review Board Statement:** Ethical approval was obtained from the Ethics Committee of Istanbul Rumeli University on 1 April 2021, with the number 2224. Written informed parental consent was obtained for all participants.

**Informed Consent Statement:** Written informed consent was obtained from all subjects involved in the study.

**Data Availability Statement:** Data is provided.

**Acknowledgments:** We thank all the children and adolescents who participated in our study, and the psychologists who helped with the application and collection of the scales. We also thank the Ministry of Internal Affairs, General Directorate of Migration Management for supporting this study.

**Conflicts of Interest:** The authors declare no conflict of interest.

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
