# Peer review of "Examination of PTSD and Depression Levels and Demographic Data of Syrian Refugee Children during the Pandemic"

_psych, doi:10.3390/psych4020018_

Round 1

Reviewer 1 Report

I want to thank the authors

for the interesting and original article. 
there are a few issues that I feel the authors should address.

First, it might have made more sense of looking

at adjustment disorders if the authors wished go

examine the association between COVID 19 events and 

mental health. The icd-11 definition of adjustment disorder eveb

explicitly states events such as the ones the 

authors investigate. Below are some articles

on the use of ptsd during covid 19 research.

it would be interesting to shortly acknowledge this

in the limitation section. 

Van Overmeire R. The Methodological Problem of Identifying Criterion A Traumatic Events During the COVID-19 Era: A Commentary on Karatzias et al. (2020). J Trauma Stress. 2020 Oct;33(5):864-865. doi: 10.1002/jts.22594. Epub 2020 Oct 2. PMID: 33007131; PMCID: PMC7675711.

Van Overmeire R. Comment on Dutheil, Mondillon, and Navel (2020): the importance of adjustment disorders and resilience. Psychol Med. 2020 Sep 8:1-2. doi: 10.1017/S003329172000344X. Epub ahead of print. PMID: 32895089; PMCID: PMC7487745.

Husky MM, Pietrzak RH, Marx BP, Mazure CM. Research on Posttraumatic Stress Disorder in the Context of the COVID-19 Pandemic: A Review of Methods and Implications in General Population Samples. Chronic Stress (Thousand Oaks). 2021 Nov 2;5:24705470211051327. doi: 10.1177/24705470211051327.
PMID: 34765850; PMCID: PMC8576091.

Second, the internal consistency of the depression scale is really low.

is this not limiting your conclusions based on this scale? 
do you have an idea why this scale is so low? 

Third, be consistent. ‘%’ is sometimes put 

before the numbers. COVID 19 is sometimes

written as Covid 19. 
line 90 states ‘trauma levels’, despite that you

are investigating ptsd levels, not trauma levels.

a trauma is an event, and not a mental health 

symptom. Therefore, change it to ptsd levels.

avoid terms as ‘effects’ and ‘impact’ throughout,

as these imply causality.

fourth, for the ptsd levels, have you looked

at only those who have experienced a trauma?

because trauma exposure is the inclusion 

criterion for ptsd. Thus, someone has to have 

experienced a trauma before we can investigate

ptsd levels. See the earlier references I mentioned 

for some information on this, if need be. 
thus, have the authors investigated for ptsd 

only those with the proper inclusion? 
if not, please mention it at least in the limitations.

Some writing errors. Eg line 245 ‘conduct’ instead of conducted.

line 264: you mention anxiety symptoms as

being in line with your study, but you have not

studied anxiety symptoms. Ptsd is a stress

related disorder. So, I would rephrase it.

overall: great article! But you should really

mention the few methodological issues I have mentioned here.

Author Response

Dear reviewer ,

Thank you for your valuable comment for the article.
Firstly, We are very sorry to report that: After your valuable comments for CDI, we just noticed
there was a problem about one of the CDI items. When we are converting from excel to spss, the
item was converted in a wrong way. So we just corrected this item and then recalculated all the
analysis. We corrected the values related CDI in the article. We are very sorry again and thank
your understanding in advance.
Regarding statistical analytical comments
We used McDonald's Omega values for CDI and CATS. Also we calculated model fit
indices of CATS and CDI. The values for reliability and validity of CATS are in acceptable
limits.
Secondly, you suggest the Kolmogorov Simirnov test for the normality of the data. Since the
sample was large enough for paremetric test, we added a reference regarding that if the sample
size consists of larger than 200 participants , parametric tests can be used.

Related to multicolienarity of the data, VIF values of the regression analysis were under two. But
in order to keep the article simple, we didnt add to the article. But if you suggest, we can add all
multicolienarity result to the statistical result.
The R2 values added to the article. B and R2 can be found in the Table 4.
The references was updated by adding new literature.
For the tables, we added it to the system.

Reviewer 2 Report

Thank you for the opportunity to review the manuscript entitled, "Examination of PTSD and Depression Levels and Demographic Data of Syrian Refugee Children during the Pandemic”. I believe this study investigated a topic relevant to the readers of “Psych”. 

The paper addresses important aspects of factors associated with symptom severities of depression and PTSD among Syrian refugee children and adolescents: the difficulties they experienced in access to food and education, the changes in their family income… The study involved a large sample of 631. The authors obtain some important results and formulate discussions of interest.

In general, this paper is well written and follows well accepted standards of academic writing. However, major revisions may prove beneficial.

The study lacks Sophisticated and/or ambitious a statistical analysis (ANOVA and Regression Analysis).

To review this article, I need to see the tables.

Introduction:

The introduction does not analyze in detail the variables under study and the relationship between variables. The discussion seems to contain information (studies) that would have been more appropriate in the Introduction.

Instruments:

The reliability of CATS - Child and Adolescent Trauma Screen Youth Report- and of Child Depression Inventory was evaluated using Cronbach’s alpha measure. Cronbach's Alpha is conditioned by the number of items and the number of alternative responses, it is necessary to use other alternative reliability indices, such as Composite Reliability (CR) and McDonald's Omega (Ω), which are calculated through factorial loads and are measured more accurate reliability. Furthermore, it is also necessary to estimate the convergent validity using the Extracted Mean Variance (AVE).

The Child Depression Inventory not present adequate reliability. Alpha reliabilities in the current study were 0.68. Values equal to or greater than .80 are considered acceptable indices of internal consistency.

The instruments must always display two important qualities: reliability and validity. It is good practice to perform a confirmatory factor analysis.  Is need to assess the validity of the constructs of the scales: Child and Adolescent Trauma Screen Youth Report and Child Depression Inventory (Confirmatory Factor Analysis: Relative Chi-Square, P; IFI; GFI; AGFI; CFI; RMSEA…)

Results:

The ANOVA analysis requires key assumptions: independent variables, univariate normality (Test Kolmogorov-Smirnov) and homoscedasticity, the assumption of equality of variances (Test Levene).

A linear regression analysis (LRA) was applied. LRA requires key assumptions: linear relationship, multivariate normality, homoscedasticity and finally, the linear regression assumes that there is little or no multicollinearity in the data. Multicollinearity occurs when the independent variables are too highly correlated with each other. Multicollinearity may be tested with three central criteria: Correlation matrix, Tolerance and Variance Inflation Factor (VIF).

What is the explained variance (R2) in the models?

Is need the values of β.

Limitations:

Finally, the authors should consider another limitations:

The alpha reliabilities in the Child Depression Inventory is low.

References:

Lack of up-to-date references.

Thank you.

Author Response

(The authors gave the same response as above.)

Reviewer 3 Report

This is a study aiming to examine the PTSD and depression levels of Syrian refugee children and adolescents. The paper is well-written; however, several changes are recommended.

In the abstract section there is not a results section. What the authors classified as "conclusions" is the main explanation of results. 

The section (in the abstract) about Key Practitioner message is not within the structured abstract format. Is this really the conclusions section of the abstract? 

In the study design, of the methods section, the authors reported that this is a cross sectional study. Further explanation would be expected.

More details are needed with regard to the variables included in sociodemographic data.

Why have the authors selected this scale? I would add a brief explanation. Why is the CATS scale better than other scales?

In the 3.4. subsection: Results concerning PTSD and depression should be separated. In the discussion section, both results should be integrated for discussion.

The conclusions and acknowledgment section should be separated from the discussion.

Author Response

All the notes that need to be reviewed are corrected and added to the manuscript.

Round 2

Reviewer 2 Report

Thank you for the opportunity to review again the manuscript entitled, "Examination of PTSD and Depression Levels and Demographic Data of Syrian Refugee Children during the Pandemic”.

The authors took into account some of my comments and they proceeded with the necessary revisions. I thank the authors their time and effort. My opinion is that in the current form the manuscript needs major revisions.

Introduction:

The introduction does not analyze in detail the variables under study and the relationship between variables. The discussion seems to contain information (studies) that would have been more appropriate in the Introduction.

Instruments:

The Child Depression Inventory not present adequate reliability. Alpha reliabilities in the current study were 0.72. Values equal to or greater than .80 are considered acceptable indices of internal consistency.

The instruments must always display two important qualities: reliability and validity.

“The confirmatory factor analyses were conducted in AMOS. The model exhibited good fit indices (x2/SD=3.15, RMSEA=.068, CFI=.71,)”

These values (RMSEA=.068, CFI=.71,) are not good.

Results:

Why ANCOVA test?

Where are the ANCOVA test results? Table 3?

Independent of the number of participants, the ANCOVA and ANOVA analysis requires the assumption of equality of variances (Test Levene).

Table 4. The values of R2 and F are incorrect. There is an R2 and F value per model.  What are the t values?

 Limitations:

Finally, the authors should consider another limitations:

The alpha reliabilities in the Child Depression Inventory is low and the model does not have a good fit.

Thank you.

Author Response

Dear reviewer,

Thank you for your valuable comments and critics about the paper. 

Here are our answers..

Introduction:

The introduction does not analyze in detail the variables under study and the relationship between variables. The discussion seems to contain information (studies) that would have been more appropriate in the Introduction.

We did some adjustments and re-writings in this part.

Instruments:

The Child Depression Inventory not present adequate reliability. Alpha reliabilities in the current study were 0.72. Values equal to or greater than .80 are considered acceptable indices of internal consistency.

The instruments must always display two important qualities: reliability and validity.

“The confirmatory factor analyses were conducted in AMOS. The model exhibited good fit indices (x2/SD=3.15, RMSEA=.068, CFI=.71,)”

These values (RMSEA=.068, CFI=.71,) are not good.

We corrected this part. There are some references that say RMSEA below .08 could be acceptable. We also add them and of course we mentioned this problem in the limitations part.

Results:

Why ANCOVA test?

Where are the ANCOVA test results? Table 3? We prefered ANCOVA test to correct the analysis for gender. But we realized the assumption was not met. So we changed it with t test for independent groups

Independent of the number of participants, the ANCOVA and ANOVA analysis requires the assumption of equality of variances (Test Levene).

Table 4. The values of R2 and F are incorrect. There is an R2 and F value per model.  What are the t values? All of them are separate regression analyses. We also add the t values.

 Limitations:

Finally, the authors should consider another limitations:

The alpha reliabilities in the Child Depression Inventory is low and the model does not have a good fit.

We revised this part 

Round 3

Reviewer 2 Report

Thank you for the opportunity to review again the manuscript entitled, "Examination of PTSD and Depression Levels and Demographic Data of Syrian Refugee Children during the Pandemic”. 
The authors took into account my comments and they proceeded with the necessary revisions. I thank the authors their time and effort. My opinion is that in the current form the manuscript can be published.